# Preferential Multi-Objective Bayesian Optimization

**Raul Astudillo**[1]            *rastudil@caltech.edu*
**Kejun Li**[1]              *kli5@caltech.edu*
**Maegan Tucker**[2]         *mtucker@gatech.edu*
**Chu Xin Cheng**[1]         *ccheng2@caltech.edu*
**Aaron D. Ames**[1]          *ames@caltech.edu*
**Yisong Yue**[1]            *yyue@caltech.edu*
[1] *California Institute of Technology*
[2] *Georgia Institute of Technology*

**Reviewed on OpenReview:** *https://openreview.net/forum?id=mjsoESaWDH*

## Abstract

Preferential Bayesian optimization (PBO) is a framework for optimizing a decision-maker's latent preferences over available design choices. While real-world problems often involve multiple conflicting objectives, existing PBO methods assume that preferences can be encoded by a single objective function. For instance, in the customization of robotic assistive devices, technicians aim to maximize user comfort while minimizing energy consumption to extend battery life. Likewise, in autonomous driving policy design, stakeholders must evaluate safety and performance trade-offs before committing to a policy. To bridge this gap, we introduce the first framework for PBO with multiple objectives. Within this framework, we propose *dueling scalarized Thompson sampling (DSTS)*, a multi-objective generalization of the popular dueling Thompson sampling algorithm, which may also be of independent interest beyond our setting. We evaluate DSTS across four synthetic test functions and two simulated tasks—exoskeleton personalization and driving policy design—demonstrating that it outperforms several benchmarks. Finally, we prove that DSTS is asymptotically consistent. Along the way, we provide, to our knowledge, the first convergence guarantee for dueling Thompson sampling in single-objective PBO.

## 1 Introduction

Bayesian optimization (BO) is a framework for optimizing objective functions with expensive or time-consuming evaluations. It has been successful in real-world applications such as hyperparameter tuning of machine learning algorithms (Snoek et al., 2012), e-commerce platform design (Letham & Bakshy, 2019), and materials design (Zhang et al., 2020). Preferential Bayesian optimization (PBO), a subframework within BO, focuses on settings where the objective function is *latent*, i.e., where the objective values cannot be observed and instead, only ordinal preference feedback from a decision-maker (DM) is observed.

While prior work in PBO has demonstrated success in various applications (Brochu et al., 2010; Nielsen et al., 2015; Tucker et al., 2020b), existing methods operate under the assumption that preferences can be encoded by a single objective function. In practice, however, problems are often characterized by multiple conflicting objectives. This occurs, for instance, when multiple users with conflicting preferences collaborate in a joint design task, as illustrated in Figure 1, or when a user wishes to explore the trade-offs between multiple conflicting attributes before committing to a design.

To motivate the need for multi-objective PBO, we examine two illustrative applications. The first application involves an exoskeleton customization task that aims to enhance user comfort. In this situation, a user assisted by an exoskeleton experiences different gait designs and indicates the most comfortable option (Tucker et al., 2020a;b). In this and other robotic assistive personalization applications, users and clinical technicians often collaborate on a design task to maximize user comfort (the user's objective) while optimizing energy

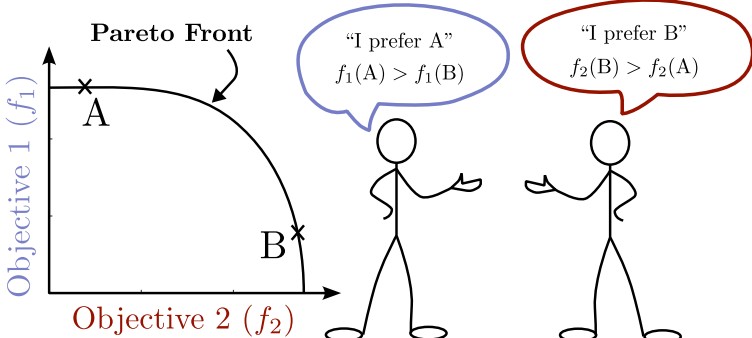

Figure 1: In this work, we extend preferential Bayesian optimization to the multi-objective setting. In contrast with existing approaches, our approach allows the decision-makers involved in the joint design task to efficiently explore optimal trade-offs between the conflicting objectives.

consumption and other metrics related to the exoskeleton's long-term functionality (the technician's objective) (Kerdraon et al., 2021).

The second application is autonomous driving policy design, where a user is presented with multiple simulations of an autonomous vehicle under different driving policies, and the user indicates the one with better safety and performance attributes (Bıyık et al., 2019). In such settings, policy-makers often seek to understand the trade-offs between multiple latent objectives, such as lane keeping and speed tracking, before committing to a specific policy (Bhatia et al., 2020).

Motivated by the applications described above, we propose a framework for PBO with multiple objectives. Our contributions are as follows:

- To the best of our knowledge, our work proposes the first framework for preferential Bayesian optimization with multiple objectives.

- We present *dueling scalarized Thompson sampling (DSTS)*, the first extension of dueling Thompson sampling (DTS) algorithms (Sui et al., 2017; Novoseller et al., 2020; Siivola et al., 2021) to the multi-objective setting.

- We prove that DSTS is asymptotically consistent. Furthermore, we also provide the first convergence guarantee for DTS in single-objective PBO.

- We demonstrate our framework across six test problems, including simulated exoskeleton personalization and autonomous driving policy design tasks. Our results show that DSTS can efficiently explore the objectives' Pareto front using preference feedback.

## 2 Related work

### 2.1 Preference-based optimization

Preference-based optimization has been actively studied across various frameworks, including multi-armed bandits (Yue et al., 2012; Bengs et al., 2021), reinforcement learning (Wirth et al., 2017), and BO (Brochu et al., 2010; González et al., 2017; Astudillo et al., 2023). It has been successful in a broad range of applications, such as personalized medicine (Nielsen et al., 2015; Sui et al., 2017; Tucker et al., 2020a), robot control (Bıyık et al., 2019; Tucker et al., 2021; Maccarini et al., 2022; Csomay-Shanklin et al., 2022) and, more recently, the alignment of large language models (Rafailov et al., 2023).

Most work in this area focuses on the single-objective setting. Two notable exceptions are the works of (Bhatia et al., 2020) and (Zhou et al., 2023). Bhatia et al. (2020) considers one-shot preference-based optimization

across multiple criteria over a finite design space. This study adopts a game-theoretic viewpoint and introduces the concept of a Blackwell winner, which implicitly requires the user to specify an acceptable trade-off between criteria, in contrast with our work. (Zhou et al., 2023) considers multi-objective preference alignment of large language models. Like our work, these two works are motivated by the idea that preference-based optimization across multiple objectives is crucial for capturing richer human feedback.

Our work extends the dueling Thompson sampling algorithm for dueling bandits introduced by (Sui et al., 2017) (termed self-sparring), which has been adapted to preference-based reinforcement learning (termed dueling posterior sampling) (Novoseller et al., 2020) and PBO (termed batch Thompson sampling) (Siivola et al., 2021). To our knowledge, we provide the first multi-objective generalization of this algorithm.

## 2.2 Multi-objective optimization

The field of multi-objective optimization has been extensively studied, encompassing both theoretical advancements and applications across various engineering problems (Miettinen, 1999; Marler & Arora, 2004; Deb, 2013). Literature within the BO framework is most closely related to our work (Khan et al., 2002; Knowles, 2006; Belakaria et al., 2019; Paria et al., 2020; Daulton et al., 2020).

Our algorithm draws inspiration from ParEGO (Knowles, 2006), a multi-objective BO algorithm that employs augmented Chebyshev scalarizations to convert a multi-objective optimization problem into multiple single-objective problems. Unlike Knowles (2006), our objectives are not observable, preventing direct modeling of scalarized values. Instead, we model each objective separately and scalarize samples drawn from these models, similar to Daulton et al. (2020)'s version of ParEGO.

Additionally, our work is related to research that incorporates user preferences into multi-objective optimization—a topic that has been actively studied both within and beyond the BO framework (Branke & Deb, 2005; Wang et al., 2017; Hakanen & Knowles, 2017; Lin et al., 2022). In most of this prior work, all objectives are assumed to be directly observable, and user preferences are captured through a latent utility function that combines these objectives into a single score to guide optimization. In contrast, we do not assume access to the objective values. Instead, we receive binary preference feedback for each objective individually, without ever observing their actual values or requiring a predefined utility function to aggregate them.

## 2.3 Additional related work

Emerging from the operations research community, the field of multi-criteria decision analysis (MCDA) focuses on decision-making under multiple conflicting criteria (Keeney & Raiffa, 1993; Pomerol & Barba-Romero, 2000). Although our work is related to this field, it diverges from the traditional MCDA approaches, which often involve aggregating preferences across criteria into a single performance measure (Young, 1974; Dyer & Sarin, 1979; Baskin & Krishnamurthi, 2009; Bhatia et al., 2020). Such aggregation requires additional assumptions about the DM's desired trade-off. Additionally, methods in this field have been explored outside the PBO framework, making them not directly applicable in our setting.

# 3 Problem setting

**Preferences** Let $\mathbb{X}$ denote the space of designs. We assume there is a DM (which may represent one or multiple users collaborating on a design task) aiming to maximize their preferences over designs. We assume the DM's preferences can be encoded via $m$ objective functions $f_1, \ldots, f_m : \mathbb{X} \to \mathbb{R}$ so that, for any given pair of designs $x, x' \in \mathbb{X}$, the DM prefers $x$ over $x'$ with respect to objective $j$ if and only if $f_j(x) > f_j(x')$. For simplicity, we assume all $m$ objectives are latent, but our approach can be easily adapted to settings where some objectives are observable, as discussed in Section 4.

**Goal** Let $f = [f_1, \ldots, f_m] : \mathbb{X} \to \mathbb{R}^m$ denote the concatenation of the $m$ objective functions. The DM seeks to find designs that maximize each objective. This concept is formalized through the notion Pareto-dominance. For a pair of designs $x, x' \in \mathbb{X}$, $x$ Pareto-dominates $x'$, denoted by $x \succ_f x'$, if $f_j(x) \geq f_j(x')$ for $j = 1, \ldots, m$ with strict inequality for at least one index $j$. The DM seeks to find the Pareto-optimal set of $f$, defined by

$\mathbb{X}_f^* := \{x : \nexists\, x' \text{ such that } x' \succ_f x\}$. The set $\mathbb{Y}_f^* := \{f(x) : x \in \mathbb{X}_f^*\}$ is termed the Pareto front of $f$. Figure 2 depicts the Pareto front for one of our test problems; the light grey region is the set of feasible objective vectors, i.e., $\{f(x) : x \in \mathbb{X}\}$ and the dark grey curve indicates the Pareto front of $f$.

**Feedback**  To assist the DM's goal, our algorithm collects preference feedback interactively (Algorithm 1). At each iteration, denoted by $n = 1, \ldots, N$, the algorithm selects a *query* constituted of $q$ designs $X_n = (x_{n,1}, \ldots, x_{n,q}) \in \mathbb{X}^q$. The DM then indicates their most preferred design among these $q$ designs for each objective. Let $r_{j,n} \in \{1, \ldots, q\}$ denote the DM's preferred design with respect to objective $j$. The collection of these responses is denoted by $r_n = [r_{1,n}, \ldots, r_{m,n}]$.

---

**Algorithm 1** Dueling Scalarized Thompson Sampling

**Input** Initial dataset: $\mathcal{D}_0$, and prior on $f$: $p_0$.
  **for** $n = 1, \ldots, N$ **do**
    Compute $p_n$, the posterior on $f$ given $\mathcal{D}_{n-1}$
    Sample $\tilde{\theta}_n$ uniformly at random over $\Theta$
    Draw samples $\tilde{f}_{n,1}, \ldots \tilde{f}_{n,q} \overset{\text{iid}}{\sim} p_n$
    Find $x_{n,i} \in \arg\max_{x \in \mathbb{X}} s(\tilde{f}_{n,i}(x); \tilde{\theta}_n),\ i = 1, \ldots, q$
    Set $X_n = (x_{n,1}, \ldots, x_{n,q})$, and observe $r_n$
    Update dataset $\mathcal{D}_n = \mathcal{D}_{n-1} \cup \{(X_n, r_n)\}$
  **end for**

---

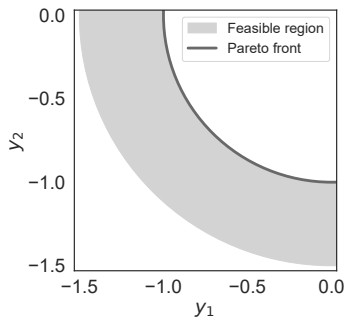

Figure 2: Feasible region and Pareto front of the DTLZ2 test function.

# 4 Dueling scalarized Thompson sampling

We introduce a novel algorithm termed *dueling scalarized Thompson sampling (DSTS)*, summarized in Algorithm 1. DSTS is obtained by adeptly combining ideas from preference-based and multi-objective optimization to derive a sound algorithm with strong performance and convergence guarantees. As is common in BO, our algorithm is comprised of a probabilistic model of the objective functions for predictions and uncertainty reasoning, along with a sampling policy that, informed by the probabilistic model, iteratively selects new queries, balancing exploration and exploitation.

## 4.1 Probabilistic model

The probabilistic model is encoded by a prior distribution over $f$, denoted by $p_0$. We assume $p_0$ consists of a set of independent Gaussian processes, each corresponding to an objective. However, our framework does not rely on this choice and can easily accommodate other priors as long as samples from the posterior distribution can be drawn.

As is standard in the PBO literature (González et al., 2017; Nguyen et al., 2021; Astudillo et al., 2023), we account for noise in the DM's responses by using a Logistic likelihood for each objective $j = 1, \ldots, m$ of the following form:

$$\mathbf{P}\left(r_{j,n} = i \mid f_j(X_n)\right) = \frac{\exp(f_j(x_{n,i})/\lambda_j)}{\sum_{i'=1}^{q} \exp(f_j(x_{n,i'})/\lambda_j)},\ i = 1, \ldots, q, \tag{1}$$

where $\lambda_j > 0$ is the noise-level parameter. We estimate $\lambda_j$ along with the other hyperparameters via maximum likelihood. We assume noise is independent across objectives and interactions.

Let $\mathcal{D}_0$ denote the initial dataset and $\mathcal{D}_{n-1} = \mathcal{D}_0 \cup \{(X_k, r_k)\}_{k=1}^{n-1}$ denote the data available right before the $n$-th interaction with the DM. Let $p_n$ denote he posterior over $f$ given $\mathcal{D}_{n-1}$. The posterior cannot be computed in closed form but can be approximated using, e.g., a variational inducing point approach (Nguyen et al., 2021). For observable objectives, the above model can be replaced by a standard Gaussian process model with a Gaussian likelihood (see Appendix B.1).

## 4.2 Sampling policy

Our primary algorithmic contribution is our sampling policy, which extends the dueling Thompson sampling (DTS) algorithmic family to the multi-objective setting. This is achieved by leveraging augmented Chebyshev scalarizations, a technique from multi-objective optimization used to decompose a multi-objective optimization problem into multiple single-objective problems. We next explain augmented Chebyshev scalarizations and describe how we integrate them with DTS.

**Augmented Chebyshev scalarizations**  Augmented Chebyshev scalarizations are widely used for multi-objective optimization (Miettinen, 1999). In BO, in particular, they were employed by Knowles (2006) and Paria et al. (2020). We also leverage them to derive a sound sampling policy in our setting.

For a given vector of scalarization parameters, $\theta \in \Theta := \{\theta \in \mathbb{R}^m : \sum_{j=1}^m \theta_j = 1 \text{ and } \theta_j \geq 0, \ j = 1, \dots, m\}$, the augmented Chebyshev scalarization function is defined by

$$s(y; \theta) = \min_{j=1,\dots,m} \{\theta_j y_j\} + \rho \sum_{j=1}^m \theta_j y_j, \tag{2}$$

where $\rho$ is a small positive constant. It can be shown that any solution of $\max_{x \in \mathbb{X}} s(f(x); \theta)$ lies in the Pareto-optimal set of $f$. Conversely, if $\rho$ is small enough, every point in the Pareto-optimal set of $f$ is a solution of $\max_{x \in \mathbb{X}} s(f(x); \theta)$ for some $\theta \in \Theta$ (Theorem 3.4.6, Miettinen, 1999).

**Dueling scalarized Thompson sampling**  At each iteration, $n$, we draw a sample from the scalarization parameters uniformly at random over $\Theta$, denoted by $\tilde{\theta}_n$. We also draw $q$ independent samples, denoted by $\tilde{f}_{n,1}, \dots, \tilde{f}_{n,q}$, from the posterior distribution on $f$ given $\mathcal{D}_{n-1}$. The next query is then given by $X_n = (x_{n,1}, \dots, x_{n,q})$, where

$$x_{n,i} \in \underset{x \in \mathbb{X}}{\operatorname{argmax}} \, s\left(\tilde{f}_{n,i}(x); \tilde{\theta}_n\right), \ i = 1, \dots, q. \tag{3}$$

Intuitively, our sampling policy operates by first determining a subset of the Pareto-optimal set of $f$ using $\tilde{\theta}_n$, denoted as $\mathbb{X}^*_{f;\tilde{\theta}_n} = \operatorname{argmax}_{x \in \mathbb{X}} s(f(x); \tilde{\theta}_n)$. Then, each $x_{n,i}$ is sampled according to the probability (induced by the posterior on $f$) that $x_{n,i} \in \mathbb{X}^*_{f;\tilde{\theta}_n}$, analogous to single-objective dueling posterior sampling (Sui et al., 2017). The DM's responses provide information of the highest value point among $x_{n,1}, \dots, x_{n,q}$ for each objective, which in turn allows us to learn about $\mathbb{X}^*_{f;\tilde{\theta}_n}$. Since $\tilde{\theta}_n$ is drawn independently at each iteration, we explore a diverse collection of subsets $\mathbb{X}^*_{f;\tilde{\theta}_n}$ within $\mathbb{X}^*_f$.

We note that our sampling policy is agnostic to the choice of the probabilistic model, provided that samples from the posterior can be drawn. In addition, our sampling policy is suitable for problems with mixed latent and observable objectives thanks to its dual interpretation as a policy for preference-based optimization (Sui et al., 2017) and traditional optimization with observable objectives (Paria et al., 2020). Specifically, when all objectives are observable, our sampling policy can be interpreted as a batch generalization (Kandasamy et al., 2018) of the scalarized Thompson sampling algorithm proposed by Paria et al. (2020).

## 4.3 Theoretical analysis

We now study the convergence properties of DSTS. We begin by analyzing the single-objective setting and establish the asymptotic consistency of DTS. To our knowledge, this is the first such result for DTS in PBO. The result is stated in Theorem 1, with a proof—based on a martingale argument—provided in Appendix A.2.

**Theorem 1.** *Suppose that $\mathbb{X}$ is finite, $m = 1$, and the sequence of queries $\{X_n\}_{n=1}^\infty$ is chosen according to the DTS policy. Then, for each $x \in \mathbb{X}$, $\lim_{n \to \infty} \mathbf{P}_n(x \in \operatorname{argmax}_{x' \in \mathbb{X}} f(x')) = \mathbf{1}\{x \in \operatorname{argmax}_{x' \in \mathbb{X}} f(x')\}$ almost surely for $f$ drawn from the prior.*

Extending this result to the multi-objective setting requires a minor modification to the DSTS algorithm. Specifically, our proof introduces a small probability of comparing against a fixed reference design in each iteration. This modification is required by our proof due to the non-linear nature of Chebyshev scalarizations

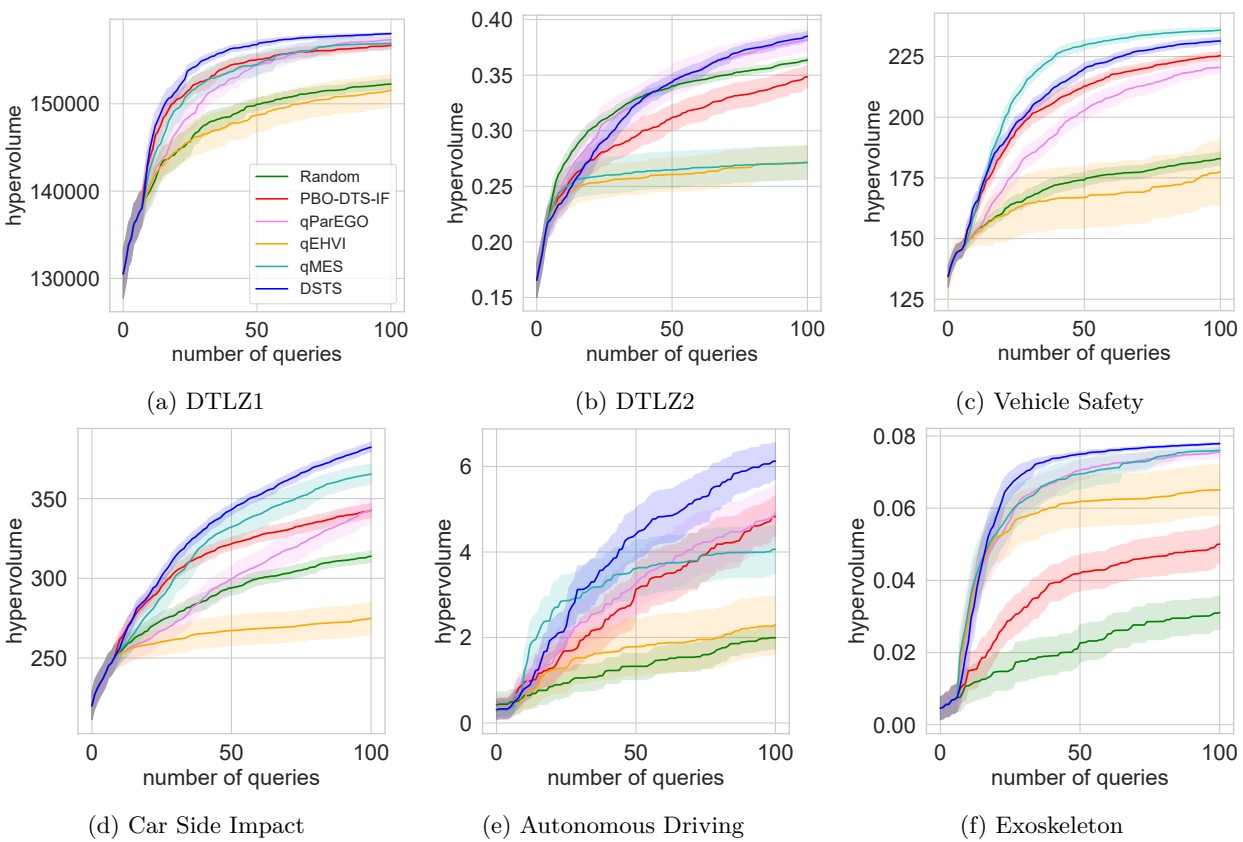

Figure 3: Our framework was demonstrated on six test problems: DTLZ1 (a), DTLZ2 (b), Vehicle Safety (c), Car Side Impact (d), Autonomous Driving (e), and Exoskeleton (f). Overall, our proposed method (DSTS) delivers the best performance. qMES and qParEGO exhibit a mixed performance, achieving good results in some test problems and poor results in others. The remaining methods, Random, PBO-DTS-IF, and qEHVI, consistently underperform DSTS.

and is not required in the single-objective case. The resulting convergence guarantee is stated in Theorem 2. The proof—which can be found in Appendix A.3—again relies on a martingale argument and the fact that varying $\theta$ allows Chebyshev scalarizations to recover all Pareto-optimal points.

Before stating the result, we describe the modified DSTS policy under consideration. Assume $q = 2$, and fix any reference point $x_{\text{ref}} \in \mathbb{X}$ and $\delta \in (0, 1)$. At each iteration, the first design $x_{n,1}$ is selected as in Equation 3, while the second design $x_{n,2}$ is set to $x_{\text{ref}}$ with probability $\delta$, or otherwise selected via Equation 3.

**Theorem 2.** *Suppose that $\mathbb{X}$ is finite, $q = 2$, and the sequence of queries $\{X_n\}_{n=1}^{\infty}$ is chosen according to the modified DSTS policy described above. Then, for each $x \in \mathbb{X}$, $\lim_{n \to \infty} \mathbf{P}_n(x \in \mathbb{X}_f^*) = \mathbf{1}\{x \in \mathbb{X}_f^*\}$ almost surely for $f$ drawn from the prior.*

We now place our results in context with prior theoretical work on DTS. Sui et al. (2017) showed that DTS achieves asymptotic consistency and sublinear regret in the dueling bandits setting, assuming independent pairs of arms. However, their analysis does not extend to our setting, where arms may be correlated. Notably, the analysis of Sui et al. (2017) relies on showing that all arms are chosen infinitely often, which may not be true in our context. Similarly, Novoseller et al. (2020) showed analogous convergence results in a reinforcement learning setting under a Bayesian linear reward model. In contrast, our result holds for non-linear objectives. Moreover, the result of Novoseller et al. (2020) also relies on showing that each arm is selected infinitely often. Finally, note that the results of Sui et al. (2017) and Novoseller et al. (2020) are only applicable in the single-objective setting; as discussed above, the multi-objective setting presents additional challenges.

Unlike prior work, we do not establish regret bounds for DSTS. Indeed, such bounds remain an open question even for DTS in single-objective PBO. While we see this as a valuable research direction, such analysis is beyond the scope of our work which primarily aims to introduce multi-objective PBO. Finally, it is important to recognize that the asymptotic consistency of data-driven algorithms like DSTS cannot be taken for granted. For instance, Astudillo et al. (2023) showed that the adaptation of qEI proposed by Siivola et al. (2021) is not asymptotically consistent and can perform poorly in single-objective PBO, despite being one of the most widely used algorithms. In Theorem 3, we show that qEHVI (Daulton et al., 2020), a multi-objective generalization of qEI, suffers from the same limitation in our setting. A proof is provided in Appendix A.4. Our empirical results support this finding, showing that qEHVI can perform very poorly.

**Theorem 3.** *There exists a problem instance with finite $\mathbb{X}$ and $q = 2$ such that if $X_n \in \text{argmax}_{X \in \mathbb{X}^q} \text{qEHVI}_n(X)$ for all $n$, then $\lim_{n \to \infty} \mathbf{P}_n(x \in \mathbb{X}_f^*) = t$ almost surely for some fixed $x \in \mathbb{X}$ and $t \in (0, 1)$.*

## 5 Numerical experiments

We evaluate our algorithm across six test problems and compare it with five other sampling policies. All algorithms are implemented using BoTorch (Balandat et al., 2020). Details on the performance metric, the benchmark sampling policies, and the test problems are provided below. The code for reproducing our experiments can be found at `https://github.com/RaulAstudillo06/PMBO`.

### 5.1 Performance metric

We quantify performance using the hypervolume indicator, which has been shown to result in good coverage of Pareto fronts when maximized (Zitzler et al., 2003). Let $\widehat{\mathbb{Y}}^* = \{y_\ell\}_{\ell=1}^L$ be a finite approximation of the Pareto front of $f$. Its hypervolume is given by $\text{HV}(\widehat{\mathbb{Y}}^*, r) = \mu\left(\bigcup_{\ell=1}^L [r, y_\ell]\right)$, where $r \in \mathbb{R}^m$ is a reference vector, $\mu$ denotes the Lebesgue measure over $\mathbb{R}^m$, and $[r, y_\ell]$ denotes the hyper-rectangle bounded by the vertices $r$ and $y_\ell$. We report performance by setting $\widehat{\mathbb{Y}}^*$ equal to the set of Pareto optimal points across designs shown to the DM.

### 5.2 Benchmarks

We compare our algorithm (DSTS) against uniform random sampling (Random), three adapted algorithms from standard multi-objective BO (qParEgo, qEHVI, qMES), and a standard PBO algorithm with inconsistent overall preference feedback (PBO-DTS-IF). Our experiments in this section use the regular version of DSTS. In Appendix B.3 we show that the modified version of DSTS used in Theorem 2 achieves virtually the same performance for small values of $\delta$. All algorithms use the same priors, and the resulting posteriors are approximated via the variational inducing point approach proposed by Nguyen et al. (2021). Approximate samples from the posterior distribution used by DSTS and PBO-DTS-IF are obtained via 1000 random Fourier features (Rahimi & Recht, 2007).

**Adapted standard multi-objective BO methods** A common approach in the PBO literature is to use a *batch* acquisition function designed for parallel BO with observable objectives, ignoring the fact that preference feedback is observed rather than objective values (Siivola et al., 2021; Takeno et al., 2023). Despite lacking the principled interpretations they enjoy in their original setting, they often deliver strong empirical performance. Following this principle, we adopt three batch acquisition functions from standard multi-objective BO as benchmarks: qParEGO (Knowles, 2006; Daulton et al., 2020), qEHVI (Daulton et al., 2020), and qMES (Belakaria et al., 2019). Since these algorithms were not originally designed for latent objectives, they require minor adaptations that we describe in Appendix B.5. These algorithms use the same probabilistic model as DSTS. Thus, any difference in performance is solely due to the use of different sampling policies.

**Single-objective PBO with inconsistent aggregated preference feedback** Single-objective PBO methods are often applied to problems characterized by multiple conflicting objectives. In such cases, DMs are expected to aggregate their preferences across objectives, which can be challenging for DMs and often

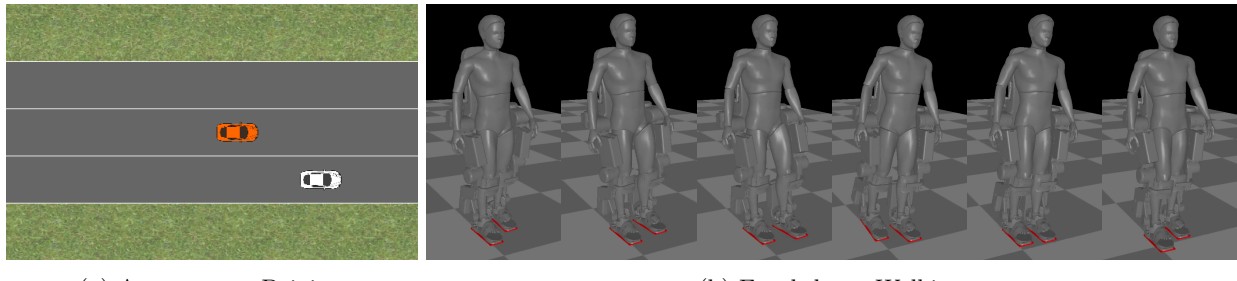

(a) Autonomous Driving  (b) Exoskeleton Walking

Figure 4: Simulation environments used in our test problems.

results in inconsistent feedback. For example, in the context of exoskeleton personalization, this would require forcing the exoskeleton user and clinical technician to reach a unified response at every iteration, which can be challenging if the user's objective is to maximize comfort while the technician's objective is to ensure the exoskeleton's long-term energy efficiency. To understand the effect of using this approach, we include a standard single-objective PBO approach using inconsistent feedback. Additional details on this benchmark are provided in Appendix B.5.

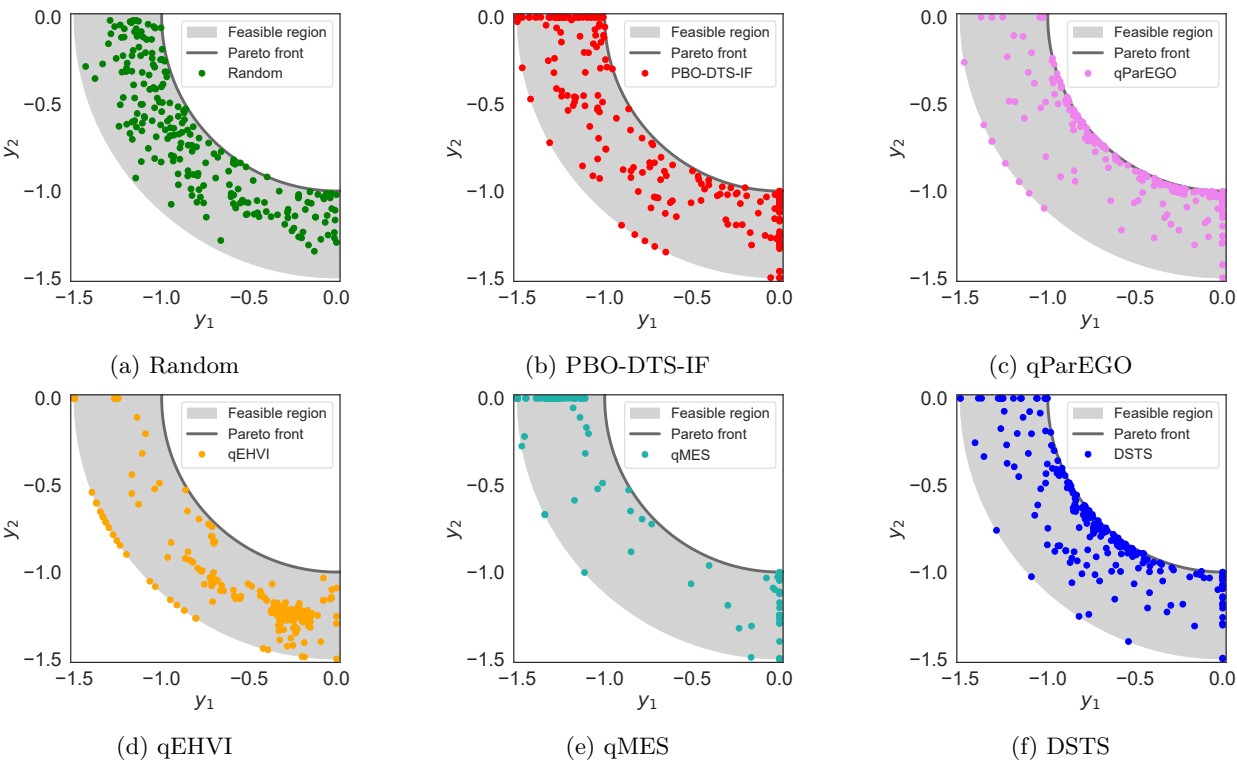

(a) Random  (b) PBO-DTS-IF  (c) qParEGO

(d) qEHVI  (e) qMES  (f) DSTS

Figure 5: Illustration of sampled designs for the DTLZ2 test function. These figures show that our proposed method (DSTS) provides a better exploration of the Pareto front than its competitors.

## 5.3 Test problems

We report performance across four synthetic test problems (DTLZ1, DTLZ2, Vehicle Safety, and Car Side Impact), a simulated autonomous driving policy design task (Autonomous Driving), and a simulated exoskeleton gait design task (Exoskeleton) using queries with $q = 2$ and $q = 4$ designs. Details of these test problems are provided below. In all problems, an initial dataset is obtained using $2(d + 1)$ queries chosen

uniformly at random over $\mathbb{X}^q$, where $d$ is the input dimension of the problem. After this initial stage, each algorithm was used to select 100 additional queries sequentially. Results for $q = 2$ are shown in Figure 3. Each plot shows the mean of the hypervolume of the designs included in queries thus far, plus and minus 1.96 times the standard error. Each experiment was replicated 30 times using different initial datasets. In all problems, the DM's responses are corrupted by moderate levels of Gumbel noise, which is consistent with the use of a Logistic likelihood (see Appendix B.2 for the details). Results for $q = 4$ can be found in Appendix B.4.

**DTLZ1 and DTLZ2**  The *DTLZ1* and *DTLZ2* functions are standard test problems from the multi-objective optimization literature (Deb et al., 2005). In our experiments, we configure DTLZ1 with $d = 6$ design variables and $m = 2$ objectives, and DTLZ2 with $d = 3$ design variables and $m = 2$ objectives. Results for these problems are shown in Figures 3(a) and 3(b), respectively. Our approach achieves the best performance in both problems, tied with qParEGO on DTLZ2.

Surprisingly, on the DTLZ2 problem, PBO-DTS-IF, qEHVI, and qMES underperform significantly, even being surpassed by Random. To understand this, we plot a representative set of objective vectors corresponding to the queried designs in Figure 5. As illustrated, Random offers a reasonable exploration of the Pareto front (likely due to the low dimensionality of DTLZ2). However, it exposes the user to many low-quality designs, which can potentially frustrate DMs. PBO-DTS-IF and qMES tend to favor designs where one of the objectives achieves its maximum possible value, which may be problematic for DMs seeking more balanced solutions. qEHVI fails to explore the Pareto front, concentrating its queries on a limited sub-optimal region instead. Finally, DSTS and qParEGO provide a more comprehensive exploration of the Pareto front.

**Vehicle Safety and Car Side Impact**  The *Vehicle Safety* and *Car Side Impact* test functions are designed to emulate various metrics of interest in the context of crashworthiness vehicle design. Overall, these test problems emulate an expert's assessment based on expensive experiments where cars are intentionally crashed, and safety metrics are evaluated. Vehicle Safety has $d = 5$ design variables and $m = 3$ objectives. Car Side Impact has $d = 7$ design variables and $m = 4$ objectives. For further details, we refer the reader to Tanabe & Ishibuchi (2020). Results for the Vehicle Safety and Car Side Impact experiments can be found in Figures 3(c) and 3(d), respectively. For the Vehicle Safety problem, qMES is the best-performing algorithm, followed by DSTS. For the Car Side Impact, DSTS performs the best, followed closely by qMES.

**Autonomous Driving Policy Design**  To supplement the synthetic test functions, we further evaluate our algorithm on a simulated autonomous driving policy design task. For this problem, we use a modification of the Driver environment presented in Bıyık et al. (2019). A similar environment was also used by Bhatia et al. (2020), providing empirical evidence that user preferences in this context are inherently governed by multiple latent objectives. In our modified environment, illustrated in Figure 4(a), an autonomous control policy is created to drive a trailing (red) vehicle forward to a goal location while maintaining some minimum distance with a leading (white) vehicle. The control policy switches between two modes, collision avoidance and goal-following, based on a minimum distance threshold. The behavior of the leading car is fixed by setting a pre-specified set of actions.

Using this simulation environment, we consider four objectives representing approximations of subjective notions of safety and performance: lane keeping, speed tracking, heading angle, and collision avoidance. The design space is parameterized by four control variables: two parameters that account for how fast the vehicle approaches the goal or the other vehicle, respectively, one position gain that accounts for the adjustment on the desired heading, and the minimum distance threshold used to switch between the two modes. The results of this experiment are shown in Figure 3(e). As illustrated, our approach again delivers better performance than its competitors.

**Exoskeleton Gait Customization**  Lastly, we evaluate our algorithm on an exoskeleton gait personalization task using a high-fidelity simulator of the lower-body exoskeleton Atalante (Kerdraon et al., 2021), illustrated in Figure 4(b). This problem emulates the scenario discussed in the introduction, in which there are two conflicting objectives: subjective user comfort and energy efficiency. For simulation purposes, we approximate comfort as a linear combination of three attributes: average walking speed (faster speed is preferred),

maximum pelvis acceleration (lower peak acceleration is preferred), and the center of mass tracking error (lower error is preferred). We approximate total energy consumption as the $l^2$-norm of joint-level torques, averaged over the simulation duration. We note that this is an observable objective. Thus, our approach is modified as discussed in Section 4 and further elaborated on Appendix B.1 to leverage direct observations of this objective.

The design space is parameterized by five gait features: step length, minimum center of mass position with respect to stance foot in sagittal and coronal plane, minimum foot clearance, and the percentage of the gait cycle at which minimum foot clearance is enforced. Each unique set of features corresponds to a unique gait. These gaits are synthesized using the FROST toolbox (Hereid & Ames, 2017) and are simulated in Mujoco to obtain the corresponding objectives. Since simulations are time-consuming, we build surrogate objectives by fitting a (regular) Gaussian process to the objectives obtained from 1000 simulations, with each set of gait features drawn uniformly over the design space. As shown in Figure 3(f), DSTS achieves the best performance, followed closely by qParEGO and qMES.

## 5.4 Discussion

Across the broad range of problems considered, DSTS delivers the best overall performance. Specifically, DSTS yields the highest hypervolume in nearly all problems (except for the Vehicle Safety problem, where it is second to qMES). Two of the standard multi-objective benchmarks, qParEGO and qMES, exhibit mixed results, highlighting the importance of developing algorithms designed to handle preference feedback as opposed to naively adapting algorithms intended for observable objectives. Notably, qEHVI is the worst-performing algorithm, even surpassed by Random. This is consistent with Theorem 3, which shows that qEHVI is not consistent in general, thus highlighting the value of our asymptotic consistency result for DSTS (Theorem 2). Lastly, PBO-DTS-IF consistently underperforms DSTS, confirming that a single-objective PBO approach is insufficient to explore the optimal trade-offs in problems with multiple conflicting objectives. The runtimes of all methods are discussed in Appendix B.6.

# 6 Conclusion

In this work, we proposed a framework for PBO with multiple latent objectives, where the goal is to help DMs efficiently explore the objectives' Pareto front guided by preference feedback. Within this framework, we introduced dueling scalarized Thompson sampling (DSTS), which, to our knowledge, is the first approach for PBO with multiple objectives. Our experiments demonstrate that DSTS provides significantly better exploration of the Pareto front than several benchmarks across six test problems, including simulated autonomous driving policy design and exoskeleton gait customization tasks. Moreover, we showed that DSTS is asymptotically consistent, providing the first convergence result for dueling Thompson sampling in PBO.

While our work provides a sound approach to tackling important applications not covered by existing methods, there are also a few limitations that suggest avenues for future exploration. Future work could include a deeper theoretical analysis of DSTS, such as investigating convergence rates and regret bounds, as well as the development of alternative sampling policies. For example, Astudillo et al. (2023) provided an efficient approach to approximate a one-step lookahead Bayes optimal policy in single-objective PBO, demonstrating superior performance against various established benchmarks. Although their approach cannot be easily adapted to our context, exploring alternative mechanisms for computing non-myopic sampling policies in our setting would be valuable. Finally, it would be interesting to explore DSTS in other settings, such as preference-based reinforcement learning.

**Acknowledgments**

This research was partially supported by Wandercraft.

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

# A    Proofs of theoretical results

## A.1    Notation and auxiliary results

We first introduce the following notation. Let $\mathcal{F}_n$ denote the $\sigma$-algebra generated by $\mathcal{D}_{n-1}$ and $\mathcal{F}_\infty$ denote the minimal $\sigma$-algebra generated by $\{\mathcal{F}_n\}_{n=1}^\infty$. We denote the conditional probability measures induced by $\mathcal{F}_n$ and $\mathcal{F}_\infty$ by $\mathbf{P}_n$ and $\mathbf{P}_\infty$, respectively. Unless otherwise stated, throughout this analysis, we assume that $f$ is a random function drawn from the prior.

We will now prove two lemmas. Lemma 1 guarantees that for any Pareto optimal point, there is a set of scalarizations with positive measure for which this point is optimal. Lemma 2, on the other hand, ensures that promising points are compared against the reference point sufficiently many times.

**Lemma 1.** *Let $f$ be any fixed function. If $x \in \mathbb{X}_f^*$, then there exists $\Psi \subset \Theta$ such that $x$ is the only element of $\operatorname{argmax}_{x' \in \mathbb{X}} s(f(x'); \psi)$ for all $\psi \in \Psi$ and $\mathbf{P}(\theta \in \Psi) > 0$, where $\theta$ is a uniform random variable over $\Theta$.*

*Proof.* From Theorem 3.4.6 in Miettinen (1999), we know that $x \in \mathbb{X}_f^*$ ( i.e. $x$ is properly Pareto optimal) if and only if there exists $\theta \in \Theta$ such that $x$ is the only element of $\operatorname{argmax}_{x' \in \mathbb{X}} s(f(x'); \theta)$. As a result, since $x \in \mathbb{X}_f^*$, there exists at least one $\theta \in \Theta$ such that $x$ is the unique maximizer of $s(f(x'); \theta)$. Since $\mathbb{X}$ is finite, the function $s(f(x'); \theta)$ is continuous with respect to $\theta$ and $x$ is the unique maximizer at $\theta$, by continuity, there exists a neighborhood around $\theta$ where $x$ remains the unique maximizer. Let $\Psi$ be this neighborhood around $\theta$. Since $\theta$ is uniformly distributed over $\Theta$, the set $\Psi$, being non-empty, has positive measure. Hence $\mathbf{P}(\theta \in \Psi) > 0$. $\square$

**Lemma 2.** *Suppose that the assumptions of Theorem 2 hold and let $x \in \mathbb{X}$ be any point with $\mathbf{P}_\infty(x \in \mathbb{X}_f^*) > 0$. Then, $f_j(x) - f(x_{\mathrm{ref}})$ is $\mathcal{F}_\infty$-measurable for $j = 1, \ldots, m$.*

*Proof.* Making a slight abuse of notation, we write $x = \operatorname{argmax}_{x' \in \mathbb{X}} s(f(x'); \theta)$ if $x$ is the only element of $\operatorname{argmax}_{x' \in \mathbb{X}} s(f(x'); \theta)$.

Let $\theta$ denote a uniform random variable over $\Theta$ independent of $\mathcal{F}_\infty$. The existence of such a $\theta$ is guaranteed by Kolmogorov's extension theorem.

From Lemma 1, there exists a $\Psi \subset \Theta$ (depending on $f$) such that $\mathbf{P}_\infty(x = \operatorname{argmax}_{x' \in \mathbb{X}} s(f(x'); \psi) \, \forall \, \psi \in \Psi) > 0$ and $\mathbf{P}_\infty(\theta \in \Psi) > 0$.

A standard Martingale argument shows that

$$\lim_n \mathbf{P}_n(x = \operatorname*{argmax}_{x' \in \mathbb{X}} s(f(x'); \psi) \, \forall \, \psi \in \Psi) = \mathbf{P}_\infty(x = \operatorname*{argmax}_{x' \in \mathbb{X}} s(f(x'); \psi) \, \forall \, \psi \in \Psi) > 0$$

almost surely.

This convergence to a positive limit implies that for sufficiently large $n$, the conditional probability is bounded below by some $\epsilon_1 > 0$ such that $\mathbf{P}_n(x = \operatorname{argmax}_{x' \in \mathbb{X}} s(f(x'); \psi) \, \forall \, \psi \in \Psi) > \epsilon_1$. Similarly, we can find $\epsilon_2$ such that $\mathbf{P}_n(\theta \in \Psi) > \epsilon_2$ for all $n$ large enough.

Under the modified DSTS policy where the $x_{n,2}$ defaults to $x_{\mathrm{ref}}$ with probability $\delta$, we have

$$\mathbf{P}_n(X_n = (x, x_{\mathrm{ref}})) \geq \mathbf{P}_n(x = \operatorname*{argmax}_{x' \in \mathbb{X}} s(f(x'); \psi) \, \forall \, \psi \in \Psi) \mathbf{P}_n(\theta \in \Psi) \delta > \epsilon_1 \epsilon_2 \delta$$

for all $n$ large enough.

It follows that the event $X_n = (x, x_{\mathrm{ref}})$ occurs for infinitely many $n$. Let $n_1, n_2, \ldots$ denote the sequence of indices such that $X_{n_k} = (x, x_{\mathrm{ref}})$. Since $q = 2$,

$$\mathbf{E}[r_{j,n_k} - 1 \mid f_j] = \mathbf{P}(r_{j,n_k} = 2)$$

$$= \frac{\exp(f_j(x_{\mathrm{ref}})/\lambda_j)}{\exp(f_j(x)/\lambda_j) + \exp(f_j(x_{\mathrm{ref}})/\lambda_j)}.$$

Thus, by the law of large numbers we have

$$\lim_{K \to \infty} \frac{1}{K} \sum_{k=1}^{K} (r_{j,n_k} - 1) = \frac{\exp(f_j(x_{\text{ref}})/\lambda_j)}{\exp(f_j(x)/\lambda_j) + \exp(f_j(x_{\text{ref}})/\lambda_j)}$$

$$= \frac{1}{1 + \exp((f_j(x) - f_j(x_{\text{ref}}))/\lambda_j)}$$

almost surely. We deduce from this that $f_j(x) - f_j(x_{\text{ref}})$ is $\mathcal{F}_\infty$-measurable for $j = 1, \dots, m$. $\qquad\square$

## A.2 Proof of Theorem 1

**Theorem 1.** *Suppose that $\mathbb{X}$ is finite, $m = 1$, and the sequence of queries $\{X_n\}_{n=1}^{\infty}$ is chosen according to the DTS policy. Then, for each $x \in \mathbb{X}$, $\lim_{n \to \infty} \mathbf{P}_n(x \in \arg\max_{x' \in \mathbb{X}} f(x')) = \mathbf{1}\{x \in \arg\max_{x' \in \mathbb{X}} f(x')\}$ almost surely for $f$ drawn from the prior.*

*Proof.* Observe that $\mathbb{X}_f^* = \arg\max_{x' \in \mathbb{X}} f(x')$ in the single-objective setting. We will show that the event $0 < \mathbf{P}_\infty(x \in \mathbb{X}_f^*) < 1$ occurs with probability zero by showing that a contradiction arises almost surely.

Since $\mathbf{P}_\infty(x \in \mathbb{X}_f^*) < 1$, there must exist a fixed $x' \in \mathbb{X}$ such that $\mathbf{P}_\infty(f(x') > f(x)) > 0$. Moreover, we can choose $x'$ such that $\mathbf{P}_\infty(x' \in \mathbb{X}_f^*) > 0$. Similar to the proof of Lemma 2, this implies that we can find $\epsilon_x, \epsilon'_x > 0$ such that $\mathbf{P}_n(x \in \mathbb{X}_f^*) > \epsilon_x$ and $\mathbf{P}_n(x' \in \mathbb{X}_f^*) > \epsilon_{x'}$ for all $n$ large enough.

Now observe that under the DTS policy,

$$\mathbf{P}_n(X_n = (x, x', \dots, x')) = \mathbf{P}_n(x \in \mathbb{X}_f^*)\mathbf{P}_n(x' \in \mathbb{X}_f^*)^{q-1} > \epsilon_x \epsilon_{x'}^{q-1}$$

for all $n$ large enough (note that we have used that $x_{n,1}, \dots, x_{n,q}$ are chosen independently). This implies that the event $X_n = (x, x', \dots, x')$ occurs infinitely often almost surely. Similar to the proof of Lemma 2, this implies that

$$\frac{\exp(f(x)/\lambda)}{\exp(f(x)/\lambda) + (q-1)\exp(f(x')/\lambda_j)}$$

and

$$\frac{\exp(f(x')/\lambda)}{\exp(f(x)/\lambda) + (q-1)\exp(f(x')/\lambda_j)}$$

are both $\mathcal{F}_\infty$-measurable. By taking the ratio between these two quantities, we see that $\exp((f(x) - f(x'))/\lambda)$ is $\mathcal{F}_\infty$-measurable, which in turn implies that $f(x) - f(x')$ is also $\mathcal{F}_\infty$-measurable. From this, it follows that $\mathbf{P}_\infty(f(x') > f(x)) = \mathbf{1}\{f(x') > f(x)\}$.

Now recall that $\mathbf{P}_\infty(f(x') > f(x)) > 0$. Thus, it must be the case that $\mathbf{P}_\infty(f(x') > f(x)) = 1$. However, this implies that $\mathbf{P}_\infty(x \in \mathbb{X}_f^*) = 0$, which is a contradiction.

From the above, it follows that $\mathbf{P}_\infty(x \in \mathbb{X}_f^*)$ is 0 or 1 almost surely. Similar to the proof of Theorem 2, we conclude that $\mathbf{P}_\infty(x \in \mathbb{X}_f^*) = \mathbf{1}\{x \in \mathbb{X}_f^*\}$ by virtue of Doob's Bayesian consistency theorem (Doob, 1949). $\qquad\square$

## A.3 Proof of Theorem 2

**Theorem 2.** *Suppose that $\mathbb{X}$ is finite, $q = 2$, and the sequence of queries $\{X_n\}_{n=1}^{\infty}$ is chosen according to the modified DSTS policy. Then, for each $x \in \mathbb{X}$, $\lim_{n \to \infty} \mathbf{P}_n(x \in \mathbb{X}_f^*) = \mathbf{1}\{x \in \mathbb{X}_f^*\}$ almost surely for $f$ drawn from the prior.*

*Proof.* A standard martingale argument shows that $\lim_{n \to \infty} \mathbf{P}_n(x \in \mathbb{X}_f^*) = \mathbf{P}_\infty(x \in \mathbb{X}_f^*)$ almost surely. Thus, it remains to show that $\mathbf{P}_\infty(x \in \mathbb{X}_f^*) = \mathbf{1}\{x \in \mathbb{X}_f^*\}$ almost surely.

To prove this, we will show that the event $0 < \mathbf{P}_\infty(x \in \mathbb{X}_f^*) < 1$ occurs with probability zero. For the sake of contradiction, assume this event holds with positive probability. As we will see next, this yields a contradiction that holds almost surely.

Since $\mathbf{P}_\infty(x \in \mathbb{X}_f^*) < 1$, there must exist $x' \in \mathbb{X}$ such that $\mathbf{P}_\infty(x' \succ_f x) > 0$. Moreover, we can choose $x'$ such that $\mathbf{P}_\infty(x' \in \mathbb{X}_f^*) > 0$.

Given $\mathbf{P}_\infty(x \in \mathbb{X}_f^*) > 0$, $f_j(x) - f_j(x_{\text{ref}})$ is $\mathcal{F}_\infty$-measurable for each $j = 1, \ldots, m$ by Lemma 2. Similarly, since $\mathbf{P}_\infty(x' \in \mathbb{X}_f^*) > 0$, $f_j(x') - f_j(x_{\text{ref}})$ is $\mathcal{F}_\infty$-measurable. Thus,

$$(f_j(x') - f_j(x_{\text{ref}})) - (f_j(x) - f_j(x_{\text{ref}})) = f_j(x') - f_j(x)$$

is $\mathcal{F}_\infty$-measurable for each $j = 1, \ldots, m$.

Next, observe that $x' \succ_f x$ if and only if $\min_{j=1,\ldots,m} f_j(x') - f_j(x) > 0$. Hence, $\mathbf{1}\{x' \succ_f x\}$ is $\mathcal{F}_\infty$-measurable and $\mathbf{P}_\infty(x' \succ_f x) = \mathbf{1}\{x' \succ_f x\}$ almost surely.

Recall that $\mathbf{P}_\infty(x' \succ_f x) > 0$. Thus, it must be the case that $\mathbf{P}_\infty(x' \succ_f x) = 1$. This implies that $\mathbf{P}_\infty(x \in \mathbb{X}_f^*) = 0$, which contradicts our initial assumption that $0 < \mathbf{P}_\infty(x \in \mathbb{X}_f^*) < 1$.

Finally, since $\mathbf{P}_\infty(x \in \mathbb{X}_f^*)$ is 0 or 1 almost surely, from Doob's Bayesian consistency theorem (Doob, 1949) we conclude that $\mathbf{P}_\infty(x \in \mathbb{X}_f^*) = \mathbf{1}\{x \in \mathbb{X}_f^*\}$. $\square$

### A.4 Proof of Theorem 3

**Theorem 3.** *There exists a problem instance with finite $\mathbb{X}$ and $q = 2$ such that if $X_n \in \text{argmax}_{X \in \mathbb{X}^q} \text{qEHVI}_n(X)$ for all $n$, then $\lim_{n \to \infty} \mathbf{P}_n(x \in \mathbb{X}_f^*) = t$ almost surely for some fixed $x \in \mathbb{X}$ and $t \in (0, 1)$.*

*Proof.* For simplicity, we focus on the single-objective case. Our example can be easily extended to the multi-objective case, e.g., by augmenting the problem with dummy constant objectives. In this case, $\mathbb{X}_f^* = \text{argmax}_{x' \in \mathbb{X}} f(x')$ and the qEHVI acquisition function reduces to the qEI acquisition function (Siivola et al., 2021), defined by

$$\text{qEI}_n(x) = \mathbf{E}_n[\{f(x) - \mu_n^*\}^+], \tag{4}$$

where $\mu_n^*$ is the maximum posterior mean value over designs previously shown.

We build on the example provided by Astudillo et al. (2023). Specifically, we let $\mathbb{X} = \{1, 2, 3, 4\}$ and consider the functions $f(\,\cdot\,; k) : \mathbb{X} \to \mathbb{R}$, for $k = 1, 2, 3, 4$, given by $f(1; k) = -1$ and $f(2; k) = 0$ for all $k$, and

$$f(x; 1) = \begin{cases} 1, & x = 3 \\ \frac{1}{2}, & x = 4 \end{cases}, \quad f(x; 2) = \begin{cases} \frac{1}{2}, & x = 3 \\ 1, & x = 4 \end{cases},$$

$$f(x; 3) = \begin{cases} -\frac{1}{2}, & x = 3 \\ -1, & x = 4 \end{cases}, \quad f(x; 4) = \begin{cases} -1, & x = 3 \\ -\frac{1}{2}, & x = 4 \end{cases}.$$

Let $s$ be a number with $0 < s < 1/3$ and set $t = 1 - s$. We consider a prior distribution on $f$ with support $\{f(\,\cdot\,; i)\}_{i=1}^4$ such that

$$p_{0,k} = \mathbf{P}(f = f(\,\cdot\,; k)) = \begin{cases} s/2, & k = 1, 2, \\ t/2, & k = 3, 4. \end{cases}$$

We assume a logistic likelihood given by

$$\mathbf{P}\left(r_n = i \mid f(X_n)\right) = \frac{\exp(f(x_{n,i})/\lambda)}{\exp(f(x_{n,1})/\lambda) + \exp(f(x_{n,2})/\lambda)}, \quad i = 1, 2.$$

From the proof of Astudillo et al. (2023), we know that $X_n = (3, 4)$ for all $n$ and the posterior distribution evolves according to the equations

$$p_{n+1,k} \propto \begin{cases} p_{n,k} w_n, & k = 1, 3, \\ p_{n,k}(1 - w_n), & k = 2, 4, \end{cases}$$

where $w_n = a\mathbf{1}\{r_n = 1\} + (1-a)\mathbf{1}\{r_n = 2\}$ and $a = \exp(1/2\lambda)$.

We will use the above to show that $p_{n,3} + p_{n,4} = t$ for all $n$. To this end, observe that the above equations imply that $p_{n,1}/p_{n,3} = p_{0,1}/p_{0,3} = s/t$ for all $n$. Similarly, $p_{n,2}/p_{n,4} = p_{0,2}/p_{0,4} = s/t$ for all $n$. Thus, $p_{n,1} = (s/t)p_{n,3}$ and $p_{n,2} = (s/t)p_{n,4}$ for all $n$. Moreover,

$$\begin{aligned}
1 &= p_{n,1} + p_{n,2} + p_{n,3} + p_{n,4} \\
&= (s/t)p_{n,3} + (s/t)p_{n,4} + p_{n,3} + p_{n,4} \\
&= (p_{n,3} + p_{n,4})(1 + s/t).
\end{aligned}$$

Therefore,

$$\begin{aligned}
p_{n,3} + p_{n,4} &= 1/(1 + s/t) \\
&= t.
\end{aligned}$$

Finally, let $x = 2$ and observe that $x \in \operatorname{argmax}_{x' \in \mathbb{X}} f(x')$ if and only if $f = f(\,\cdot\,;3)$ or $f = f(\,\cdot\,;4)$. Hence,

$$\begin{aligned}
\mathbf{P}_n\left(x \in \operatorname*{argmax}_{x' \in \mathbb{X}} f(x')\right) &= \mathbf{P}_n\left(f = f(\,\cdot\,;3) \text{ or } f = f(\,\cdot\,;4)\right) \\
&= p_{n,3} + p_{n,4} \\
&= t.
\end{aligned}$$

$\square$

# B Additional experimental details and results

## B.1 Additional details on our probabilistic model under observable objectives

Suppose for concreteness that objective $j$ is observable. Then, at each iteration $n$, we observe (potentially noisy) measurements of $f_j(x_{n,1}), \ldots, f_j(x_{n,q})$. Let $y_{n,1}, \ldots, y_{n,q}$ denote these measurements, and let $\mathcal{D}_{n-1} = \{(x_{k,i}, y_{k,i})\}_{k=1,\ldots,n,i=1,\ldots,q}$ denote all the measurements available right before the $n$-th interaction with the DM. As is standard in the BO literature, we can assume Gaussian noise such tat $y_{n,i} = f_j(x_{n,i}) + \epsilon_{n,i}$ where the terms $\epsilon_{n,i} \sim N(0, \sigma^2)$ are independent across $n$ and $i$. If we assume a Gaussian prior over $f_j$, then the posterior distribution of $f_j$ given $\mathcal{D}_{n-1}$ is again a Gaussian process, whose mean and covariance functions can be computed using the standard Gaussian process regression equations (equations 2.23 and 2.24 of Rasmussen & Williams, 2006).

## B.2 Noise in the DM's responses

In our experiments, we simulate noise in the DM's responses using additive Gumbel noise. Specifically, if $X_n$ is the query presented at iteration $n$, then the response observed is $r_n = [r_{1,n}, \ldots, r_{m,n}]$, where

$$r_{j,n} = \operatorname*{argmax}_{i=1,\ldots,q} f_j(x_{n,i}) + \epsilon_{n,i,j},$$

and $\epsilon_{n,i,j} \sim \text{Gumbel}(0, \lambda_j^{\text{true}})$ for $i = 1, \ldots, q$ and $j = 1, \ldots, m$ are independent. Under this choice, we have

$$\mathbf{P}\left(r_{j,n} = i \mid f_j(X_n)\right) = \frac{\exp(f_j(x_{n,i})/\lambda_j^{\text{true}})}{\sum_{i'=1}^{q} \exp(f_j(x_{n,i'})/\lambda_j^{\text{true}})},$$

for $i = 1, \ldots, q$, i.e., this recovers a Logistic likelihood.

Following Astudillo et al. (2023), in each problem, for every objective, $\lambda_j^{\text{true}}$ is chosen such that, on average, the DM makes a mistake 20% of the time when comparing pairs of designs among those with the top 1% objective values within $\mathbb{X}$. We estimate this percentage by uniformly sampling a large number of design points over $\mathbb{X}$.

Noise in the DM's responses observed by PBO-DTS-IF is generated by scalarizing the corrupted objective values defined above using a Chebyshev scalarization. Concretely, if $r'_n$ denotes the response observed by PBO-DTS-IF when presented with query $X_n$, then

$$r'_n = \operatorname*{argmax}_{i=1,\ldots,q} s(f(x_{n,i}) + \epsilon_{n,i}; \tilde{\theta}_n),$$

where $\epsilon_{n,i} = [\epsilon_{n,i,1}, \ldots, \epsilon_{n,i,m}]$, $\epsilon_{n,i,j}$ is defined as before, and $\tilde{\theta}_n$ is drawn uniformly from $\Theta$.

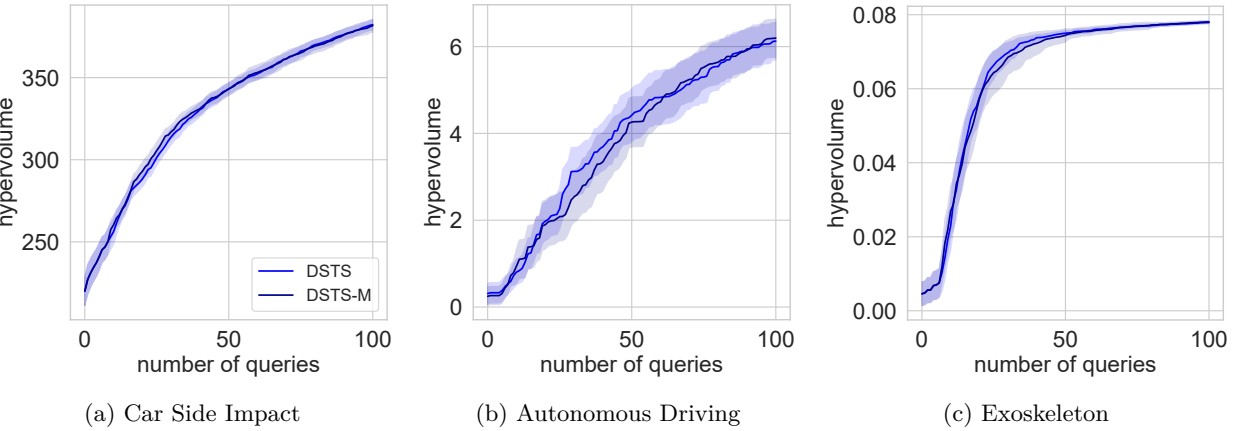

(a) Car Side Impact  (b) Autonomous Driving  (c) Exoskeleton

Figure 6: Performance of DSTS and modified DSTS (DSTS-M) using queries with $q = 2$ alternatives across three test problems. Their performance is almost identical in the three problems considered.

## B.3  Results for modified DSTS

We evaluate the performance of the modified version of DSTS used in the proof of Theorem 1 on three of our test problems. We set $x_{\mathrm{ref}}$ by drawing a point uniformly at random over $\mathbb{X}$. This point is different for each replication. In addition, we set $\delta = 0.05$ such that $x_{\mathrm{ref}}$ is selected five times on average across 100 iterations. The results are shown in Figure 6.

## B.4  Results for $q = 4$

We carry out experiments analogous to those presented in the main paper, using queries with $q = 4$ alternatives each. We focus on DSTS and the two strongest benchmarks, qParEGO and qMES. The results are shown in Figure 7. For a clearer comparison, we also include results for $q = 2$. As depicted, increasing the number of alternatives improves the performance of the three algorithms. DSTS delivers the best overall performance under queries with $q = 2$ and $q = 4$ alternatives.

## B.5  Additional details on our benchmarks

### B.5.1  Adapted standard multi-objective BO algorithms

**qParEGO**  In standard multi-objective BO with observable objectives, the qParEGO acquisition function is defined by $\alpha_n(x) = \mathbf{E}_n[\{s(f(x); \theta_n) - \max_{x' \in \mathcal{X}_n} s(f(x'); \theta_n)\}^+]$, where $\mathcal{X}_n$ is the set of all designs in queries presented to the DM up to time $n$ and $\theta_n$ is drawn uniformly at random over $\Theta$. Following Siivola et al. (2021), we replace $f(x')$ by the posterior mean of $f$ at $x'$ to avoid taking the expectation with respect to the random variable $\max_{x' \in \mathcal{X}_n} s(f(x'); \theta_n)$.

**qEHVI**  In the standard multi-objective BO setting with observable objectives, the qEHVI acquisition function is defined by $\alpha_n(x) = \mathbf{E}_n[\mathrm{HV}(\mathcal{Y}_n \cup \{f(x)\}, r) - \mathrm{HV}(\mathcal{Y}_n, r)]$, where $\mathcal{Y}_n$ is the set of objective vectors corresponding to designs presented to the DM up to time $n$. As with qParEGO, we replace these objective vectors with posterior mean vectors to avoid taking the expectation with respect to these unknown values.

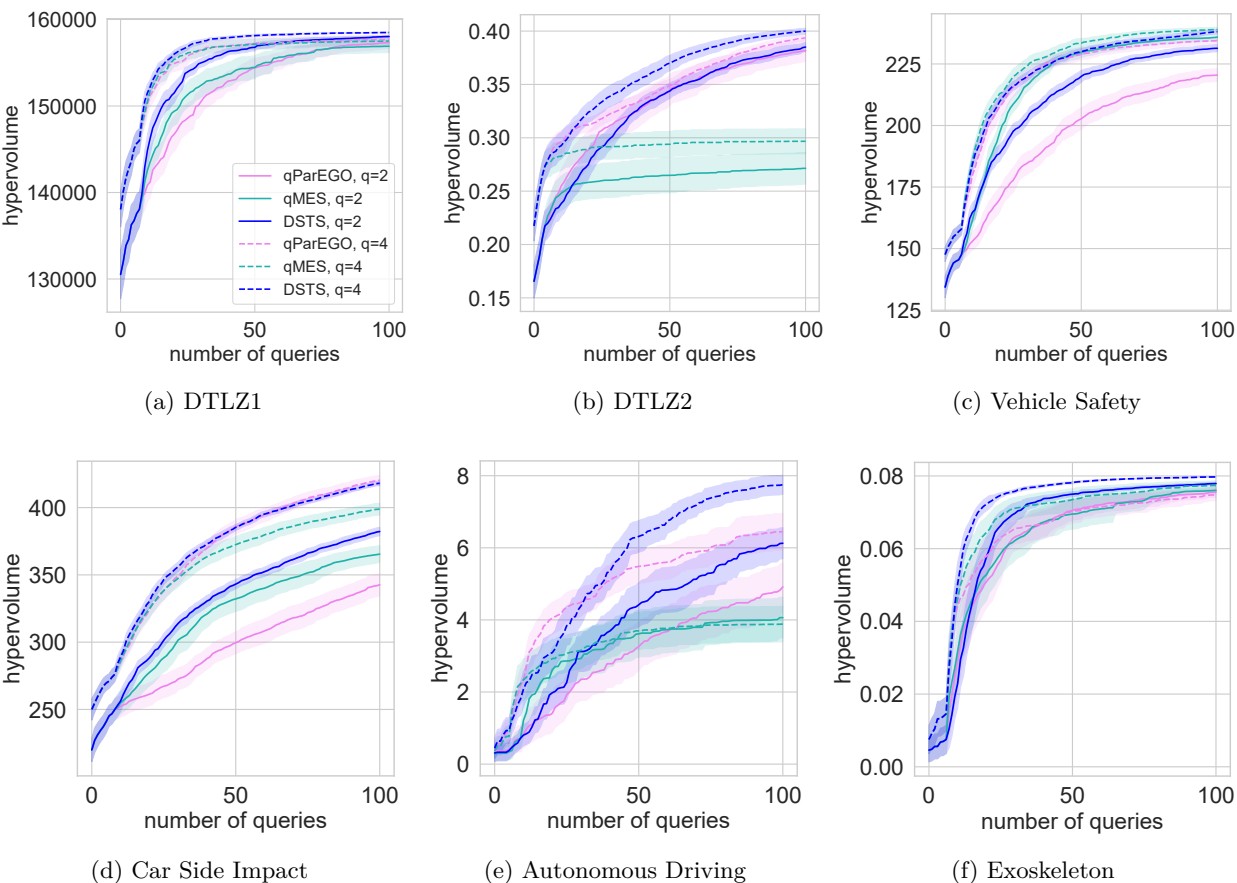

Figure 7: Performance of qParEGO, qMES and DSTS using queries with $q = 2$ and $q = 4$ alternatives across our six test problems. Using a larger number of alternatives improves the performance of the three algorithms. DSTS delivers the best overall performance under queries with $q = 2$ and $q = 4$ alternatives.

Note that qEHVI also requires specifying reference point $r$, which is challenging if the objectives are latent. In our experiments, we set $r$ equal to the coordinate-wise minimum posterior mean value across designs presented to the DM up to time $n$.

**qMES** Unlike qParEGO and qEHVI, qMES does not require any modification to be applied in our setting. However, it is worth noting that the lookahead step in the definition of qMES assumes that the objective values will be observed at the selected points. However, this is not the case in our setting. We believe this can be problematic as it may cause qMES to over-value queries constituted by designs with very distinct objective values, even though preference feedback from such queries can often be uninformative.

### B.5.2 PBO with inconsistent preference feedback

The feedback used by PBO-DTS-IF is produced as follows. At each iteration $n$, a set of scalarization parameters $\tilde{\theta}_n$, is drawn uniformly at random over $\Theta$. Given a query $X_n = (x_{n,1}, \ldots, x_{n,q})$ , we assume the DM then provides a noisy response to $\arg\max_{i=1,\ldots,q} s(f(x_{n,i}); \tilde{\theta}_n)$. Inconsistency arises from sampling different scalarization parameters at every iteration, which, in general, implies that preferences cannot be encoded by a single objective function. We argue this mimics the DM's desire to explore the trade-off between objectives before committing to a solution. At the same time, we note that these responses respect intuitive user behavior, such as preference for queries for which each objective is as large as possible.

The responses provided by the DM are used to fit a (single-output) Gaussian process with a Logistic likelihood for which posterior inference is carried out using the same approach we use for the other algorithms. New

queries are generated using the (single-objective) dueling Thompson sampling strategy under this probabilistic model. The performance of this method is expected to be poor when the Pareto-optimal set is large, i.e., when the trade-offs between objectives are significant.

| Problem/Algorithm | PBO-DTS-IF | qParEGO | qEHVI | qMES | DSTS |
|---|---|---|---|---|---|
| DTLZ1 | 5.7 | 24.3 | 25.6 | 58.5 | 19.2 |
| DTLZ2 | 6.8 | 16.4 | 10.6 | 16.5 | 13.3 |
| Vehicle Safety | 8.4 | 15.4 | 36.7 | 43.8 | 17.8 |
| Car Side Impact | 5.9 | 38.1 | 352.8 | 324.4 | 36.3 |
| Autonomous Driving | 5.5 | 34.1 | 102.7 | 72.2 | 32.1 |
| Exoskeleton | 6.3 | 16.5 | 23.7 | 59.3 | 14.6 |

Table 1: Average runtimes in seconds for all the algorithms compared (except for Random) accross all test problems.

## B.6  Runtimes

The average runtimes for all algorithms across all test problems are presented in Table 1. DSTS is comparable with qParEGO and faster than qEHVI and qMES. PBO-DTS-IF is the faster algorithm since, in contrast with the other algorithms, it requires a single Gaussian process model instead of $m$. However, it is worth noting that the runtimes of these algorithms could be reduced by parallelizing the training step of its $m$ models. Moreover, the runtimes of DSTS could be further reduced by parallelizing the generation of the $q$ alternatives in each query during the optimization step.

