# OpenReview forum: "Preferential Multi-Objective Bayesian Optimization"
_TMLR — Accepted by TMLR_

### Review · Reviewer_Ja3w · 2025-02-12

**Summary Of Contributions:**

This paper introduces a new Bayesian optimization setting where the goal is to maximizing multiple latent objectives using preference observations.

The authors propose an acquisition function called dueling scalarized Thompson sampling, which performs Thompson sampling on a scalarized objective based on an augmented Chebyshev scalarization.
They prove the asymptotic consistency of the proposed sampling policy under two conditions: (a) the batch size is fixed at \(q = 2\) and (b) the policy is modified so that, with a constant probability, one of the queries is a fixed reference point.

Experiments on six benchmark problems demonstrate that the proposed sampling policy outperforms baselines in most cases.
Since this paper explores a new setting, the baselines are primarily acquisition functions for non-preferential multi-objective Bayesian optimization, adapted to the new setting.

**Audience:**

Yes

**Claims And Evidence:**

Yes

**Requested Changes:**

1. In Section 4.3, the assumptions and conditions need be clearly stated.
Without these, it is unclear how to interpret and verify the statement.
Specifically, it is unclear what is the exact assumption on \\(f\\).
By skimming through the appendix, it seems that \\(f\\) is an unknown *fixed* function and the probability \\(\Pr_n(x \in \mathbb{X} _ f^\*)\\) is only over the stochasticity of Thompson sampling.
However, it is well-known that GPs cannot fit all functions.

1. Make baseline details more clear.
Since most baselines are acquisition functions for non-preferential BO adapted to preference observations, it is crucial to include these details in the main paper.
As TMLR is a journal without page limit, it is better to move their descriptions to the main paper.

**Strengths And Weaknesses:**

1. This paper proposes preferential multi-objective Bayesian optimization.
To the best of my knowledge, this setting has not been studied before, and seems to be useful in practice.
Hence, it is interesting to investigate this setting.

1. The sampling policy proposed in this paper (dueling scalarized Thompson sampling) is quite standard.
It is essentially Thompson sampling of the scalarized objectives, which is exactly the same as Praria et al. (2020).
In addition, this Thompson sampling policy has been used as a baseline in previous multi-objective BO papers (e.g., Daulton et al., 2020)
Thus, I do not think the sampling policy itself is a contribution of this paper.
Instead, the message from this paper is that Thompson sampling outperforms other multi-objective acquisition functions like qEHVI, qMES, and qParEGO in preferential multi-objective BO.
Those baselines were not initially designed for preferential BO, though.

---

> ### Author Response · Authors · 2025-02-27
> **Response to Reviewer Ja3w**
>
> Dear Reviewer Ja3w,
>
> Thank you for your comments and questions. We are glad you appreciate the novelty and relevance of the problem setting introduced by our work. Below, we address the points raised in your review. Please let us know if you have any further questions or comments.
>
> **Q1.** *“…I do not think the sampling policy itself is a contribution of this paper. Instead, the message from this paper is that Thompson sampling outperforms other multi-objective acquisition functions…”*
>
> **A1.** Our sampling policy, dueling scalarized Thompson sampling (DSTS), can be derived from two perspectives. The first views it as an “adaptation” of scalarized Thompson sampling (DTS) (introduced by Paria et al., 2023) to the preference-based setting, analogous to how our baselines are derived. However, this is not the perspective we take in our work. Instead, we derive DSTS as a sound generalization of DTS from the (preference-based) single-objective setting to the (preference-based) multi-objective setting. While both perspectives lead to the same sampling policy, the latter provides deeper insight into DSTS’s strong empirical performance. This perspective itself bears novelty, and thus we argue that DSTS is indeed a contribution of our work.
>
> This reasoning also raises the question: can any of our baselines be derived as a sound generalization of a single-objective PBO sampling policy? While the notion of a "sound" generalization can be subjective, we argue that this is not possible for qParEGO and qEHVI. Both are generalizations of qEI, which has been shown by Astudillo et al. (2023) to be a fundamentally flawed sampling policy in the preference-based setting.
>
> On the other hand, a sound generalization of qMES is indeed possible by modifying the lookahead step to condition on the actual observations—binary comparisons rather than objective values. However, we chose not to pursue this generalization in our work, as entropy-based acquisition functions are computationally expensive and often require approximations that degrade performance. Exploring such a generalization while addressing these challenges would be an interesting direction for future work.
>
> We hope this discussion also clarifies the distinction between DSTS and the baselines considered in our work.
>
> **Q2.** *“In Section 4.3, the assumptions and conditions need be clearly stated… it is unclear what is the exact assumption on f… and the probability Prn(x∈Xf∗)is only over the stochasticity of Thompson sampling.”*
>
> **A2.** We appreciate this observation. Theorem 1 assumes that $f$ is drawn from the (correctly specified) prior distribution. This type of result can be seen as a Bayesian counterpart to frequentist analysis, which typically assumes $f$ is a fixed function from a pre-specified function class (e.g., the reproducing kernel Hilbert space induced by a kernel function). We have revised the statement of Theorem 1 to explicitly include this assumption. We have also added additional details in the statements and proofs in Section A to ensure clarity throughout our analysis.
>
> **Q3.** *“Make baseline details more clear… it is better to move their descriptions to the main paper.”*
>
> **A3.** While we acknowledge the importance of detailing how the baselines are adapted to preference observations, we believe including these details in the main paper would divert attention from our core contributions—namely, introducing preferential multi-objective Bayesian optimization as a novel setting and establishing DSTS as a sound sampling policy for this setting. To maintain clarity and focus, we have kept these details in the appendix, where interested readers can find a thorough explanation. We hope this strikes a good balance between completeness and readability. Please let us know if you have any further concerns.

---

### Review · Reviewer_pJ9s · 2025-02-13

**Summary Of Contributions:**

The paper considers the problem of multi-objective black-box optimization where each objective can be evaluated with a preference feedback. In order to address this problem, the paper introduces a method for extending preferential Bayesian optimization (PBO) to problems with multiple latent objectives. The key idea is to model each latent objective with an independent Gaussian process and leverage Chebyshev scalarization to map the multi-objective problem into a single-objective problem. At each iteration, the algorithm (named DSTS i.e. dueling scalarized Thompson sampling) draws independent posterior samples of the objectives, applies a random scalarization, and then selects a batch of designs. It is shown theoretically that a modified version of DSTS is asymptotically consistent in the multi-objective setting. Empirically, the method is evaluated on six test problems—including four synthetic benchmarks and two simulated tasks in exoskeleton gait customization and autonomous driving policy design. The experiments suggest that DSTS outperforms several adapted baselines (such as qParEGO, qMES, qEHVI) in terms of exploring the Pareto front, as measured by the hypervolume indicator.

**Audience:**

Yes

**Broader Impact Concerns:**

N/A.

**Claims And Evidence:**

Yes

**Requested Changes:**

Please see strengths and weaknesses section.

**Strengths And Weaknesses:**

- The paper considers a new understudied problem of multiple objective black-box optimization with preference evaluations.

- The proposed approach is simple and works well on synthetic and simulation benchmarks.

- The theoretical analysis is sound. It would be good to clarify the new aspect in the theory (which seems to be Lemma 2) for the readers.

- The paper is very well written and easy to understand.

Questions

- Since it is assumed that the same decision maker provides preferences for multiple objectives, there is a higher chance for the objectives to be correlated. Please discuss or analyze the choice of using independent GPs for modeling the objectives versus using a multi-output/multi-task GP.

-  Since the evaluation metric in the experiments section is based on hypervolume, it might be useful to consider scalarization techniques that are directly to hypervolume improvement. Please see [1] for reference.

- It does seem a bit counterintuitive that larger batch size leads to faster convergence in Figure 7.

- The analysis of DTLZ2 where the proposed method doesn't perform better than the baselines is nice. However, there is limited description of why does the method performs well on other benchmarks where it outperforms other baselines.

- It is surprising that qMES is consistently better than qEHVI in all the benchmarks since EI acquisition functions tend to show better practical performance (even in single objective preference optimization [2]).

- I appreciate the two applications (exoskeleton task and autonomous driving policy design) mentioned in the paper but providing preferences for multiple objectives might also increase the cognitive burden for the decision maker. Single objective preference queries are preferred over scalar rewards primarily because of decrease in cognitive burden but I am not sure if this advantage remains with multiple objectives.


[1] Zhang, R., & Golovin, D. (2020, November). Random hypervolume scalarizations for provable multi-objective black box optimization. In International conference on machine learning (pp. 11096-11105). PMLR.
[2] Raul Astudillo, Zhiyuan Jerry Lin, Eytan Bakshy, and Peter Frazier. qEUBO: A decision-theoretic acquisition
function for preferential Bayesian optimization. In International Conference on Artificial Intelligence and
Statistics, pp. 1093–1114. PMLR, 2023.


Overall, I think the paper studies a new problem and presents a simple approach that works well. I am happy to recommend acceptance for this paper.

---

> ### Author Response · Authors · 2025-02-27
> **Response to Reviewer pJ9s**
>
> Dear Reviewer pJ9s,
>
> We sincerely appreciate your thoughtful questions and positive appraisal of our work. We are pleased that you recognize the novelty of our problem setting, the simplicity and strong empirical performance of our algorithm, and the clarity of our manuscript. Below, we address the points raised in your review. Please let us know if you have any further questions or comments.
>
> **Q1.** *“…there is a higher chance for the objectives to be correlated. Please discuss or analyze the choice of using independent GPs for modeling the objectives versus using a multi-output/multi-task GP.”*
>
> **A1.** We agree that objectives may often be correlated in real-world applications, and using a multi-output GP to capture such correlations could be beneficial—particularly when prior knowledge about the correlation structure is available and can be encoded in the covariance function. However, since our primary contributions are introducing preferential multi-objective Bayesian optimization as a new problem setting and proposing DSTS as a principled algorithm for this setting, we opted for a simpler model with independent GPs to maintain clarity and focus. Exploring more sophisticated models, including multi-output GPs, is an interesting direction for future work.
>
> **Q2.** *“…it might be useful to consider scalarization techniques that are directly to hypervolume improvement.”*
>
> **A2.** Like the reviewer, we were also interested in evaluating the performance of the scalarization proposed by Golovin and Zhang (2011). Preliminary experiments, which were not included in the paper, indicated that this scalarization performs similarly to Chebyshev scalarization. Ultimately, we chose Chebyshev scalarization because it is known to recover the entire Pareto front (Theorem 3.4.6, Miettinen, 1999). To our knowledge, no analogous theoretical guarantee exists for the hypervolume scalarization introduced by Golovin and Zhang (2011).
>
> **Q3.** *“It does seem a bit counterintuitive that larger batch size leads to faster convergence in Figure 7.”*
>
> **A3.**  We note that in our experiment, the number of iterations is held fixed. This differs from typical batch Bayesian optimization experiments, where the total number of function evaluations is fixed while varying the batch size, leading to fewer iterations for larger batch sizes. Given this setup, we find it intuitive that larger values of $q$ lead to faster convergence. Each query provides $q−1$ pairwise comparisons (relative to the preferred point in the query set), meaning that as $q$ increases, more preference information is collected per iteration, accelerating convergence.
>
> **Q4.** *“ The analysis of DTLZ2…is nice. However, there is limited description of why does the method performs well on other benchmarks where it outperforms other baselines.”*
>
> **A4.** We are glad that you found this analysis helpful. Most of the benchmarks we consider involve more than two objectives, making an analogous analysis more challenging and likely less insightful. We did conduct a similar analysis for DTLZ1, another two-objective problem, but it did not yield any additional insights.
>
> **Q5.** *“It is surprising that qMES is consistently better than qEHVI in all the benchmarks since EI acquisition functions tend to show better practical performance (even in single objective preference optimization [2])”*
>
> **A5.** We were also surprised by the poor performance of qEHVI, which even underperformed Random in some cases. We conjecture that the erratic behavior of qEI, as shown by Astudillo et al. (2023) and extended to qEHVI in our work, becomes even more problematic in the multi-objective setting. Intuitively, this may occur because qEHVI’s pressure to improve across multiple objectives leads it to more quickly select queries that are less informative—i.e., points where the user’s response is already well understood according to the posterior. As a result, the posterior changes little from one iteration to the next, potentially causing qEHVI to get stuck, as observed for DTLZ2 in Figure 5.

---

> > ### Author Response · Authors · 2025-02-27
> > **Response to Reviewer pJ9s (continued)**
> >
> > **Q6.** *“…providing preferences for multiple objectives might also increase the cognitive burden for the decision maker. Single objective preference queries are preferred over scalar rewards primarily because of decrease in cognitive burden but I am not sure if this advantage remains with multiple objectives.”*
> >
> > **A6.** We believe that in many applications, providing preferences for each objective separately may impose a lower cognitive burden than giving an aggregated preference response over all objectives. For instance, in the exoskeleton personalization task, the latter would require both the clinical technician and the exoskeleton user to agree on a unified response, which may be challenging. Intuitively, providing scalar rewards (which inherently combine preferences across objectives) would be even more difficult, and thus likely more cognitively demanding than providing separate preference responses for each individual objective.
> >
> > **Q7.** “The theoretical analysis is sound. It would be good to clarify the new aspect in the theory…”
> >
> > **A7.** We appreciate the reviewer’s positive assessment of our theoretical analysis. As noted, the key novel aspect of our result is encapsulated in Lemma 2. Intuitively, this result ensures that promising points are compared with the reference point frequently enough. To better highlight the novel aspects of our proof, we have added a comment in Section A.1 clarifying the roles of Lemmas 1 and 2.

---

> > > ### Comment · Reviewer_pJ9s · 2025-03-24
> > >
> > > I thank the authors for their response to my questions. I think the responses adequately addressed my questions. I am mostly worried that the problem motivation might be a bit contrived (same concern as Reviewer Pwf1) but I find the response reasonable and recommend accepting the paper.

---

> ### Author Response · Authors · 2025-03-27
>
> Dear Reviewer pJ9s,
>
> We are glad to hear that our response has adequately addressed your questions and that you recommend acceptance of our work. Regarding your concern about the problem motivation, we kindly refer you to our response to Reviewer Pwf1 for additional details.
>
> Thank you again for your valuable feedback and questions.

---

### Review · Reviewer_Pwf1 · 2025-02-28

**Summary Of Contributions:**

The paper presents a framework for Preferential Bayesian Optimization (PBO) in a multi-objective setting. The authors introduce Dueling Scalarized Thompson Sampling (DSTS), an extension of Dueling Thompson Sampling (DTS) to the multi-objective case. The method is evaluated on synthetic test functions and two real-world inspired applications (exoskeleton personalization and autonomous driving policy design). The authors prove the asymptotic consistency of DSTS.

**Audience:**

Yes

**Claims And Evidence:**

Yes

**Requested Changes:**

See weaknesses section.

**Strengths And Weaknesses:**

Strengths
1. The paper shows convincing empirical results that the new algorithm performs well on this new class of problems.
2. The writing is generally clear and informative.


Weaknesses
1. I somewhat feel that the problem setting is a bit contrived and would like to see more motivation for it. Do we really have a situation where a DM or a set of DMs are giving preference feedback on individual outcomes? Couldn't we ask them to give preference feedback on multiple outcomes (see next comment) instead? That seems simpler.
2. PBO has previously been viewed in past work as helping to address a “multi-objective”-like situation (see, e.g., [1], where the authors discuss this in the introduction), where the preference function is over a multi-dimensional outcome space (using the language of Figure 1, this would be (f1, f2)). The idea in [1] is that when one is in a multi-objective setting, there always exists some latent utility over the Pareto front, and PBO can be more efficient in exploring the Pareto front because it explicitly tries to learn this utility function.
In the current paper, there is one sentence dedicated to this. I wonder if the authors could do a better comparison here?
2. Related to the above comment: in this work, once the Pareto frontier has been found, what happens next? I would be curious to understand this point, especially in the context of the applications given here. For these applications, it seems like there must then be a “final decision maker” who selects a point from the Pareto frontier? Would that decision maker put us back into the framework of [1]?
3. I think the qEUBO paper [2] is not discussed in the Related Work. I think it should be added and compared.
4. “assuming independent pairs of arms. This result does not apply to our setting, where arms are may be correlated” -- can the authors clarify this point? Suppose in the new setting we had independent Gaussian priors on each arm. Could we then prove the convergence rate result in Thm 2 of [3]?
5. Figure 5 is very interesting. However, I’m not sure saying the qParEGO is worse than DSTS because of the region near y2=0 is convincing since DSTS also doesn’t explore that area fully + DSTS seems to have more non-Pareto points. Further, DSTS doesn’t explore the area y1=0 very well compared to qParEGO.
6. I find the qEHVI result interesting. Perhaps that could be moved to the main paper with some intuition.

[1] https://proceedings.mlr.press/v151/jerry-lin22a/jerry-lin22a.pdf

[2] https://proceedings.mlr.press/v206/astudillo23a.html

[3] https://arxiv.org/pdf/1705.00253

---

> ### Author Response · Authors · 2025-03-27
> **Response to Reviewer Pwf1,**
>
> Dear Reviewer Pwf1,
>
> Thank you for your comments and questions. We are glad you appreciate the clarity of our manuscript and the strength of our empirical results. Below, we address the points raised in your review. Please let us know if you have any further questions or comments.
>
> **Q1.** *“Do we really have a situation where a DM or a set of DMs are giving preference feedback on individual outcomes? Couldn't we ask them to give preference feedback on multiple outcomes (see next comment) instead?”*
>
> **A1.** The example based on the exoskeleton customization task is inspired by a real-world situation studied by the authors. In this case, we note that the problem is indeed multi-objective because the exoskeleton user wishes to maximize comfort while the technician wishes to optimize energy consumption. As discussed in the description of our benchmark based on single-objective PBO with inconsistent aggregated preference feedback (Section 5.3), forcing users to aggregate their preferences across objectives can be challenging and often results in inconsistent feedback. As shown in our numerical experiments, this can lead to poor exploration of the Pareto front. Please also see A3 for further discussion on why a thorough exploration of the Pareto front may be desirable in some situations.
>
> **Q2.** *“PBO has previously been viewed in past work as helping to address...”*
>
> **A2.** We observe that the functions  $f_1,\ldots, f_m$​ in our work should be interpreted as utility functions rather than outcome functions, as in [1]. In particular, we note that unlike in [1], the functions  $f_1,\ldots, f_m$​ are not observable in our setting. More fundamentally, unlike [1], our framework involves multiple preference responses—one for each utility function—rather than a single aggregated response. As discussed in A1, this setting arises in real-world applications and differs from that of [1] and similar works.
>
> **Q3.** “...once the Pareto frontier has been found, what happens next?”
>
> **A3.**  Typically, once an approximation of the Pareto front has been found, the decision-maker(s) analyze the trade-offs between objectives and select a solution from the approximate set. This process, referred to as post-hoc selection of Pareto-optimal solutions, is standard in the multi-objective optimization literature. As the reviewer notes, some selection strategies may require additional elicitation, but this is not always necessary. Finally, we emphasize that in high-stakes settings such as the autonomous driving policy design use case discussed in our work, a comprehensive understanding of the Pareto front is often critical before implementing a solution. In such cases, aggregating preference feedback across objectives would be inappropriate, as it would hinder a broad exploration of the Pareto front.
>
> **Q4.** *“...qEUBO paper [2] is not discussed in the Related Work. I think it should be added and compared.”*
>
> **A4.** qEUBO is mentioned in the Conclusion (though not by name). We agree that a more thorough discussion is warranted and will add a comparison to qEUBO in the Related Work section of the revised manuscript.
>
> **Q5.** *“Suppose in the new setting we had independent Gaussian priors on each arm. Could we then prove the convergence rate result in Thm 2 of [3]?”*
>
> **A5.** No, under independent Gaussian priors on each arm, the convergence rate provided in Theorem 2 of [3] is not applicable. First, observe that this result assumes a Beta-Bernoulli model, so it does not directly apply to a setting with Gaussian priors. More fundamentally, this model implicitly assumes that the prior probabilities that $x_1$ is preferred over $x_2$​ are drawn independently for each pair $(x_1, x_2)$. This assumption does not hold in our setting, where these probabilities are induced by a latent function $f$ and are therefore correlated across pairs of arms.
>
> **Q6.** *“Figure 5 is very interesting. However, I’m not sure saying the qParEGO is worse than DSTS…”*
>
> **A6.** Thank you for pointing this out. We agree that the current phrasing overstates the comparison. In fact, DSTS and ParEGO perform comparably in this instance, as also reflected in the average performance across multiple replications shown in Figure 3b. We will revise the wording to more accurately reflect this.
>
> **Q7.** *“I find the qEHVI result interesting. Perhaps that could be moved to the main paper…”*
>
> **A7.** We are pleased to hear you found the qEHVI result interesting. We agree that it adds value to the main narrative and will move the result, along with a brief discussion, to the main paper.

---

### Decision · Action_Editor_yjBn · 2025-04-07

**Recommendation:** Accept as is

**Comment:**

This manuscript concerns multi-objective Bayesian optimization in a preferential setting, where observations take the form of preferences over pairs (or more) of points in the design space rather than direct measurements of the objective functions. To this end, the authors propose a dueling algorithm based on scalarized Thompson sampling. This algorithm appears to work well in practice in an empirical study, and the authors also provide some theoretical consistency results.

Overall, the reviewers received this work well during the initial review phase, noting that the manuscript was well-written and that the ideas presented were well motivated and defended. The reviewers did ask a series of questions and raise a series of comments, which were addressed satisfactorily by the authors during the discussion phase.

Ultimately the three reviewers were unanimous in their recommendation to accept the paper to TMLR, a decision I also support.

**Audience:**

Yes, multi-objective Bayesian optimization is an important subject that interests a nontrivial number of TMLR audience members.

**Claims And Evidence:**

Yes. The reviewers agree that the claims in this manuscript are both well-motivated and supported by convincing empirical analysis.